# Post-Operative Greater Tuberosity Resorption or Malreduction Is Associated with Poor Prognostic Outcomes in Patients with Proximal Humeral Fractures Treated Operatively—A Single-Center Retrospective Cohort Study

**DOI:** 10.3390/diagnostics13172789

**Published:** 2023-08-29

**Authors:** Kuan-Yu Lu, Ting-Han Tai, Yu-Hsin Liu, Chang-Jung Chiang, El-Wui Loh, Chin-Chean Wong, Jeffrey J. Wu

**Affiliations:** 1Department of Orthopedics, Shuang Ho Hospital, Taipei Medical University, New Taipei City 23561, Taiwan; 18fnewtaipeibook2@gmail.com (K.-Y.L.); b101101132@tmu.edu.tw (T.-H.T.); dragochevalier@gmail.com (Y.-H.L.); cjchiang@tmu.edu.tw (C.-J.C.); 2Department of Orthopedics, School of Medicine, College of Medicine, Taipei Medical University, Taipei 11031, Taiwan; 3Graduate Institute of Clinical Medicine, College of Medicine, Taipei Medical University, Taipei 11031, Taiwan; lohelwui@tmu.edu.tw; 4Center for Evidence-Based Health Care, Department of Medical Research, Shuang Ho Hospital, Taipei Medical University, New Taipei City 23561, Taiwan; 5Cochrane Taiwan, Taipei Medical University, Taipei 11031, Taiwan; 6Department of Medical Imaging, Shuang Ho Hospital, Taipei Medical University, New Taipei City 23561, Taiwan; 7International PhD Program for Cell Therapy and Regenerative Medicine, College of Medicine, Taipei Medical University, Taipei 11031, Taiwan; 8Research Center of Biomedical Devices, Taipei Medical University, Taipei 11031, Taiwan

**Keywords:** greater tuberosity, resorption, malreduction, humerus, prognosis

## Abstract

(1) Background: Proximal humerus fractures can be a debilitating condition if not properly treated. These fracture patterns are varied and differ in every patient. Functional outcomes may be determined by the integrity of the shoulder girdle involving the rotator cuff insertion. The post-operative resorption or malreduction of the greater tuberosity (GT) is an important factor contributing to the poor functional outcome of a patient. Thus, we intend to evaluate the cause-and-effect relationship between GT complications and clinical prognosis and outcomes. (2) Methods: A single-center retrospective comparative study was performed to evaluate the functional outcomes of patients undergoing operative fixation for this injury. A total of 387 consecutive cases treated operatively from 2019–2021 were included for analysis. (3) Results: 94 cases fulfilled our criteria for analysis. A matched-group comparison of 19 patients each was performed to compare demographics, post-operative fracture characteristics and clinical outcomes. (4) Conclusions: The resorption or malreduction of the GT contributes greatly to the prognostic outcome in patients treated with open reduction and internal fixation (ORIF) surgery. In our demographic study, obesity is another contributing factor affecting the parameters of post-operative reduction in proximal humerus fractures. Appropriate surgical planning and post-operative multidisciplinary care must be taken into consideration to attain a satisfactory prognostic outcome.

## 1. Introduction

Proximal humeral fractures, defined as fractures occurring at or proximal to the surgical neck of the humerus, are one of the most frequent injuries affecting the shoulder girdle in adults [1]. An estimated 706,000 proximal humeral fractures occurred worldwide in 2000 [2]. In 2008, proximal humeral fractures lead to 185,000 emergency department visits in the United States alone [3]. It is an osteoporosis-related fracture, and its incidence is rising as the population ages, from 4% fifty years ago to 7% of all fractures nowadays [4,5]. In the United States, the number of patients presenting with proximal humeral fractures is expected to reach 275,000 by 2030 [3].

The vast majority of proximal humeral fractures can be treated nonoperatively [3,6]. Although most studies support good clinical outcomes of conservative treatment, a recent prospective study has shown that significant functional impairment may occur even in non-displaced proximal humeral fractures. According to the study, over two-thirds of patients had chronic pain and significant reductions in functional capacity [7]. After the introduction of locking compression plates (LCP) around the year 2000, surgical intervention has become increasingly common, which may be attributed to the angular stability of a locking screw-plate construct even in an osteoporotic bone [6,8]. Several studies have revealed high rates of healing and excellent functional recovery in patients with proximal humeral fractures treated with LCP [9,10]. Despite the high success rate of surgical treatments, implant-related complications, such as screw back out/cut out, humeral head varus collapse, subacromial impingement, avascular necrosis, nonunion, malunion and deep infection were also reported [11,12]. Some authors have reported that the unsatisfactory clinical results may be related to the lack of experience of the treating surgeon, inexact technique, and/or specific failure of the material [13,14].

As a result, determining how to minimize complications and optimize functional outcomes has emerged as the primary concern of the orthopedic surgeon. Several surgical techniques and radiographic parameters have been proposed, such as the ideal place for plate position, restoration of anatomic neck-shaft angle and medial calcar hinge support, the use of a calcar screw, suture fixation through the rotator cuff and using strut grafting when necessary [15,16,17]. Taken together, the key concept would be building a stable construct in order to neutralize the deforming force.

In clinical practice, incorporating non-absorbable suture fixation through the entheses of the rotator cuff into the plate construct has become a surgical routine. However, resorption and malreduction of the GT has occasionally occurred, despite adequate fixation being achieved intra-operatively. Miyamura et al. have studied the relationship between fragment characteristics, intraoperative factors and post-operative GT resorption in complex proximal humeral fracture [18]. However, the underlying causes and mechanisms of these complications still remain largely unknown. Thus, this study aims to evaluate the functional outcome of patients with proximal humeral fractures presented with postoperative resorption or malreduction of GT and analyze the relevant risk factors.

We hypothesize that the occurrence of post-operative GT resorption or malreduction would invariably lead to poor clinical outcomes in patients who received surgical fixation of proximal humeral fractures, as characterized both radiographically and clinically.

## 2. Materials and Methods

### 2.1. Study Setting

This is a single-center study. In this retrospective comparative study, the requirement for written informed consent by the patients was waived.

### 2.2. Patient Enrollment

We retrospectively collected data of patients diagnosed with proximal humeral fractures treated operatively from 2019 to 2021. After removing the identification information, all of the radiographs and CT images were reviewed by 3 orthopedic surgeons who were not involved in the care of these patients. The variables determined with consensus agreement of ≥2 reviewers were used. We included fractures involving the greater tuberosity defined as 11B1.1, 11C3 in the AO/OTA classification system [19] and determined whether the GT was resorbed or in malreduction in the follow-up period. The GT involvement was defined on plain film with obvious GT displacement (>5 mm), or a definite fracture line around the GT on computed tomography (CT). We excluded patients diagnosed with AO/OTA 11A fractures (2-part fractures and isolated GT fractures) and AO/OTA 11B1.2 fractures (unifocal fractures with lesser tuberosity involvement), periprosthetic fractures, open fractures, pathological fractures, proximal humeral bone tumor and patients presenting with neurovascular injury of the affected limb. We included the patients treated operatively with a LCP and excluded those treated with arthroplasty. Patients with a follow-up period of less than 24 weeks and with incomplete data were excluded.

Three hundred and eighty-seven consecutive cases with proximal humeral fractures treated operatively with LCP were initially reviewed, and 293 cases were subsequently excluded, with the criteria as abovementioned. A total of 94 patients were included in the analysis. AO Philos, Zimmer ALPS, and Aplus proximal humeral LCP were used among these patients according to the surgeons’ preference. All of the LCP have multiple side holes that allow additional suture fixation. The average length of follow-up was 45.55 weeks. We identified 19 patients (20.2%) in whom bone resorption or malreduction of the GT developed after ORIF and 75 patients (79.8%) without any radiographic detectable resorption or malreduction of the GT (Figure 1 and Figure 2). The average time to determine the development of GT resorption was 14.72 ± 8.8 weeks after surgery. The average time to determine the development of greater tuberosity malreduction was 4.2 ± 1.77 weeks after surgery.

### 2.3. Surgical Technique

Most surgeries were performed within the first week of the injury (from 1 to 29 days, on an average of 2 days). The surgeries were performed in beach-chair position by several senior surgeons or under the direct supervision of a senior surgeon. An extended deltopectoral approach with limited detachment of the anterior deltoid or an anterolateral approach through a deltoid-splitting was used according to the surgeons’ preference. The periosteal attachment of the GT and the medial hinge was preserved. The reduction was performed under intraoperative fluoroscopy and kept temporarily by the use of Kirschner wires. The rotator cuff was secured through the enthesis with several heavy nonabsorbable sutures (No. 5 Ethibond), which would later be threaded through the plate to reinforce the construct. A new proximal humeral locking compression plate was positioned at least 5 to 8 mm distal to the upper end of the GT and 2 mm lateral to the bicipital groove to prevent plate impingement. The tendon of the long head of biceps was checked to ensure that a sufficient gap was maintained from the plate. The plate was then fixed definitively with several angular stable screws into the humeral head. The humeral shaft holes were fixed with either locking screws or standard cortical screws according to the surgeons’ decision. A final check was done to ensure good reduction quality of the GT, optimal plate position and no screw penetration from the humeral head under an image intensifier before the wound closure.

### 2.4. Data Collection

Demographic information, including gender, height, weight, body mass index (BMI), smoking status and alcohol use was collected through chart review. We assessed the bone quality using the deltoid tuberosity index in the preoperative humeral AP radiograph. The location for calculating the deltoid tuberosity index is defined as directly proximal to the deltoid tuberosity where the outer cortical borders become parallel. It is calculated by dividing the outer cortical by the inner endosteal diameter at this level [20].

For fracture characteristics, the preoperative plain films or CT image were reviewed to determine the AO/OTA classification, the presence of a varus or valgus fracture type and a fracture-dislocation. Surgical characteristics were also collected which include the interval from injury to surgery and artificial bone graft use. We assessed the immediate postoperative radiographic parameters, including inadequate medial support, lateral humeral offset, head diameter, head height, perpendicular height, neck-shaft angle, superior displacement of greater tuberosity above articular surface, medial gap (whether the gap was more than 4 mm or not) and use of a calcar-specific screw. Malreduction of the GT is defined as out of 5 to 10 mm under the tangent line of the humeral head, perpendicular to the humeral shaft axis according to previous studies [21,22]. GT resorption was defined as a more than one half decrease in the diameter of GT in the follow-up period compared with the initial postoperative plain film. Inadequate medial support is defined as either no intact or anatomically reduced calcar, no stable head-on-shaft impaction or no superiorly directed oblique locking screw that was appropriately placed into the infero-medial quadrant of the proximal humeral head fragment on the immediate postoperative radiograph. For postoperative functional outcomes, the Disabilities of the Arm, Shoulder and Hand (DASH) questionnaire was used for evaluation.

Those with GT resorption or malreduction were independently matched by age and sex by the use of a propensity score to those without resorption or malreduction (*n* = 19) in a 1:1 ratio for further analysis.

### 2.5. Statistical Analysis

Continuous variables were described with the use of means, standard deviation and ranges. Categorical variables were tabulated with absolute and relative frequencies. Numerical data were studied with the Student’s t test. Categorical variables were studied with the Chi-Square test of independence. A significant difference is defined as *p* < 0.05. Because the current study was preliminary, we estimated the effect size of the final sample available instead of an estimation of the sample size required before the study using a webtool available at https://sample-size.net/ (accessed on 22 September 2022).

## 3. Results

### 3.1. Patient Demographics & Surgical Characteristics

For all 343 patients, 153 patients were excluded due to unsuitable fracture pattern, periprosthetic fracture (*n* = 1), pathological fracture (*n* = 1) and incomplete clinical data (*n* = 1). Moreover, nine patients were excluded for the non-plate fixation method (hemiarthroplasty = 8; intramedullary nail = 1). Finally, 122 patients were not included because they were followed up within less than 24 weeks. In all the 94 patients included, the mean age was 63.32 years old, with female predominance (67%, *n* = 63). The average body height was 159.16 ± 9.54 cm and the average weight was 63.94 ± 14.41 kg with average BMI of 25.09 kg/m^2^. Less than 10% of included patients were reported to have an alcohol (9.57%) or smoking (8.51%) habit. The average deltoid tuberosity index measured on plain radiograph was 1.41. The majority of patients (87%, *n* = 82) were classified as AO/OTA 11B1.1 and 12 (13%) as AO/OTA 11C3. The average interval from injury to surgery was 2.93 ± 5.4 days and the average follow up period was 45.55 weeks. Postoperative GT resorption was found in 13 patients (13.82%) and malreduction was noted in 11 patients (11.7%). We pooled them together for analysis and there were overall 19 patients noted with the event of postoperative greater tuberosity resorption or malreduction. The average period from surgery to first notice of the event was 8.77 ± 7.86 weeks (Table 1).

### 3.2. Analysis of Factors Predisposed to Greater Tuberosity Absorption or Malreduction

As shown in Table 2, the age and gender variables between patients with or without GT resorption or malreduction were similar after matching.

The demographic and fracture characteristics between the 19 patients with greater tuberosity resorption or malreduction and the matched groups were similar except for BMI. The average BMI of patients in the resorption or malreduction group (27.35 kg/m^2^) was significantly higher than patients in the matched group (23.85 kg/m^2^), indicating that obesity may play a certain role in GT-related complications.

In contrast, the surgical parameters, which include fracture pattern (Neer and AO classifications), time to surgery, utilization of bone grafts and follow up period had no influence on the occurrence of GT absorption or malreduction. Interestingly, despite 16% of patients in the GT resorption or malreduction group being found to have inadequate medial support post-operatively, the difference was not significant when compared with the controls (*p* = 0.2297).

To further analyze the correlation between radiographic outcomes and the occurrence of GT absorption or malreduction in the two groups of patients, we have compared the postoperative radiographic parameters accordingly, as shown in Table 3.

The results have demonstrated that there was no significant difference in lateral humeral offset, head diameter, head height, perpendicular height and neck-shaft angle among patients with or without GT resorption or malreduction. A total of 68.42% of patients in the GT absorption or malreduction group had their fragment displacement above the humeral articular surface. From a biomechanical viewpoint, the radiographic outcomes also highlighted the importance of reconstructing medial support during operation by minimizing the residual medial gap or through the use of a calcar-specific screw that could offer further assistance in reducing the possibility of GT complications.

Finally, the results of the post-operative functional evaluation at the final follow-up visit revealed that patients without GT absorption or malreduction (55.47 ± 27.03) displayed a significantly better functional performance than their counterparts (95.08 ± 39.79) in terms of DASH score, with *p* = 0.0053 (Figure 3).

## 4. Discussion

In the current study, we have analyzed the incidence of GT absorption or malreduction in proximal humeral fractures and their correlation with clinical prognosis and outcomes. The results showed that the presence of postoperative GT resorption or malreduction was associated with a higher DASH score, which indicated poor clinical outcomes. Moreover, obesity has been recognized as a significant risk factor which may contribute to postoperative GT resorption or malreduction. The positive relationship between low local bone mineral density measured by deltoid tuberosity index and postoperative GT resorption or malreduction was not established. From a biomechanical perspective, a larger residual medial gap (>4 mm) and the absence of calcar-specific screws were found to be highly correlated to the occurrence of postoperative GT resorption or malreduction.

The shoulder girdle is the link between the trunk and the upper extremities and is the most dynamic and mobile joint in the body. It consists of many articulations and the glenohumeral joint is no doubt the main contributor to its motion [23]. The proximal humerus, from both an anatomical and fracture perspective, is conceptualized as consisting of the humeral head, the greater and the lesser tuberosities and the humeral shaft, as first proposed by Neer in 1970 [24,25]. The GT is located laterally on the proximal humerus. It serves as the insertion site of three tendons: the supraspinatus tendon superiorly, the infraspinatus tendon posterosuperiorly, and the teres minor tendon posteroinferiorly [26]. The GT is positioned on an average of 9 mm (range from 6 to 10 mm) below the most proximal aspect of the humeral head. This head-to-tuberosity distance is extremely important to allow adequate rotator cuff function. Insufficient rotator cuff tension and subacromial impingement may occur if the GT heals excessively proximally. On the contrary, cuff strain and failure may occur if it is placed too low. It has been shown that the inability to reconstitute the normal head-tuberosity distance leads to suboptimal results in both anatomical fracture reconstruction and arthroplasty [21,22]. Clavert et al. also pointed out that the clinical outcomes were influenced by the quality of the GT, with an inferior Constant score noted in the patients with a malunion of the GT [27]. In this retrospective comparative study assessing 94 complex proximal humeral fractures that underwent ORIF, the presence of postoperative GT resorption or malreduction was associated with a higher DASH score, which indicated poor clinical outcomes.

Based on the fact that the average time to determine the development of greater tuberosity resorption was 14.72 ± 8.8 weeks after surgery and the average time to determine the development of greater tuberosity malreduction was 4.2 ± 1.77 weeks after surgery, we inferred that the cause of malreduction is early construct failure (e.g., knot loosening of the suture through the rotator cuff or tissue cut through by the suture) and the cause of resorption is avascular necrosis or atrophic nonunion despite a strong suture fixation. According to the Ministry of Health and Welfare in Taiwan, obesity is defined as BMI ≥ 27.0 and in the demographic analysis, we found that obesity is related to postoperative GT resorption or malreduction. A number of factors have been identified as affecting bone healing, and obesity is one of them [28]. There is a complex relationship between obesity and the bone. Obesity affects bone metabolism in many ways, both potentially positive and negative, and its net influence is controversial [29,30,31,32,33]. Obesity is traditionally believed to be beneficial to bone health owing to a well-established positive effect of mechanical loading conferred by body weight to stimulate bone formation by decreasing apoptosis and increasing proliferation and differentiation of osteoblasts and osteocytes [34]. Also, a greater amount of estrogen, a protecting factor against osteoporosis by reducing bone resorption and stimulating bone formation, is present in the adipose tissue [35]. Nevertheless, obesity may decrease bone formation while increasing adipogenesis because adipocyte and osteoblasts are derived from a common multi-potential mesenchymal stromal cell [36]. Moreover, obesity may increase osteoclastogenesis and bone resorption through upregulating proinflammatory cytokines, such as IL-6 and TNF-α, which are capable of stimulating osteoclast activity through the RANKL/RANK/OPG pathway [37,38,39]. Aside from its impact on bone, obesity also increases the risk of tendinopathy, enthesopathy, tendon tear and rupture and postoperative complications [40,41]. Several studies have shown that, in obese subjects, tendons frequently undergo degeneration. A higher BMI was associated with macroscopic tendon changes, such as greater thickness but lower stiffness [42,43]. The main histopathological findings are a relative paucity of small collagen fibrils, impaired remodeling process, deposition of lipid droplets which can lead to tendolipomatosis and a disorganized architecture in the tension site [44,45]. Preclinical studies also demonstrated that the tendons of rats with obesity induced by a high-fat diet had compromised biomechanical and healing properties [46,47]. The cause might be the increased mechanical stress and the low-grade, chronic inflammatory microenvironment [48]. The inflammatory status seems to be caused by adipocytes because they suffer from hypoxic, mechanical and oxidative stress secondary to cell hypertrophy, ultimately leading to cellular apoptosis, release of the intracellular pro-inflammatory molecules and continued recalling of macrophages [49,50,51,52]. The relatively poor quality of the rotator cuff tendon in an obese patient might lead to tissue cut through by the suture, leading to early malreduction of the greater tuberosity. Excessive instability, caused by the poor quality of the tendon accompanied with compromised vascularization owing to fracture and surgical intervention, might merely permit the formation of fibrous tissue and the development of an atrophic non-union [45], eventually seen on X ray as greater tuberosity resorption.

The influence of local bone mineral density (BMD) on proximal humeral fractures is controversial [53,54,55,56,57]. A cadaveric study conducted by Fankhauser et al. revealed that low local BMD negatively affects the stability of ORIF of proximal humerus [53]. Krappinger et al. also reported low local BMD to be a predictor for later failure of ORIF in their clinical study [55]. In contrast, Franz Kralinger et al. failed to establish a positive association between BMD and the rate of mechanical failures [54]. Additionally, Anil Taskesen et al. also echoed that osteoporosis is not the main factor affecting the surgical outcomes in proximal humeral fractures [56]. Mats Bue et al. also showed that osteoporosis does not affect the functional outcome after ORIF of displaced three- or four-part proximal humeral fractures in a prospective multicenter study [57]. In concern of nonunion, a cohort nested case–control study also showed that osteoporosis is not a risk factor for the development of nonunion [58]. When merely focusing on the GT of the humerus, the data are scarce. Satoshi Miyamura et al. have concluded that GT with a larger number of fragments, smaller fragments and fragments with a lower bone density have higher rates of resorption in their previous report [18]. According to the literature, local bone mineral density could be measured with methods ranging from measuring the cortical thickness on AP radiographs, dual-emission X-ray absorptiometry, or calculating BMD on peripheral quantitative CT scans [59,60,61,62]. On AP radiographs of the shoulder, the combined cortical thickness (Tingart measurement) is a frequently reported method used to measure bone quality [62]. Deltoid tuberosity index is another simple and effective method reported for measuring the bone quality with a good correlation with peripheral quantitative CT scans and Tingart measurement [20]. In our study, we also applied a similar method by measuring the deltoid tuberosity index and checked whether there was an association between postoperative GT resorption or malreduction and local bone mineral density. The results further echoed previous reports that BMD may not play major role in GT absorption or malreduction.

In our study, 16 of 19 patients with GT resorption or malreduction were found to have superior displacement of the GT above the articular surface in the immediate postoperative radiograph. Among which, 13 patients were noted with GT resorption and 11 patients had further GT displacement that met the definition of malreduction. These radiographic parameters were related to the quality of reduction and the stability of fixation that influenced the functional outcomes [63]. A larger residual medial gap normally accompanied a more comminuted medial cortex or a non-anatomical reduced medial hinge. Under those circumstances, a higher rate of varus collapse could be anticipated in the absence of calcar-specific screws, as previously reported by Dheenadhayalan et al. [63,64].

## 5. Conclusions

Post-operative GT resorption or malreduction was associated with poor prognostic outcomes in patients with proximal humeral fractures treated with ORIF. Surgeons should be committed to obtain an adequate reduction and secure fixation of the GT on a stable locking plate construct. Obesity, a patient-specific risk factor which is modifiable, should be handled through a multidisciplinary approach.

## 6. Limitations

Firstly, this is a retrospective study. In this study, the DASH score was used to assess functional outcomes. It is a questionnaire assessing patient’s function subjectively. The range of motion of the shoulders was not assessed. Secondly, due to the vast majority of the subjects being elderly patients, some comorbidities, such as cancer, senile dementia, Parkinsonism and geriatric frailty might bias the results. Further studies are needed to investigate this problem.

## Figures and Tables

**Figure 1 diagnostics-13-02789-f001:**
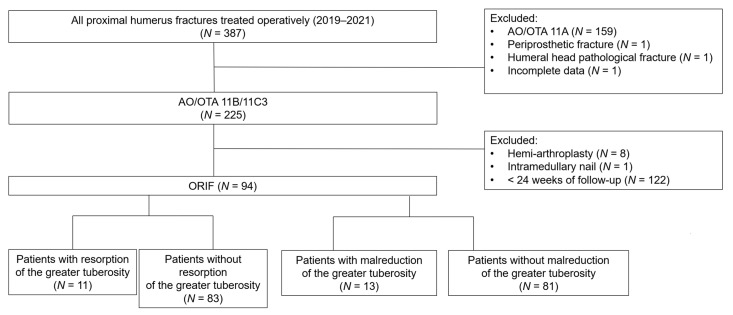
Flow chart of inclusion and exclusion criteria of the study cohort of patients with proximal humeral fractures. Abbreviation: ORIF, open reduction and internal fixation.

**Figure 2 diagnostics-13-02789-f002:**
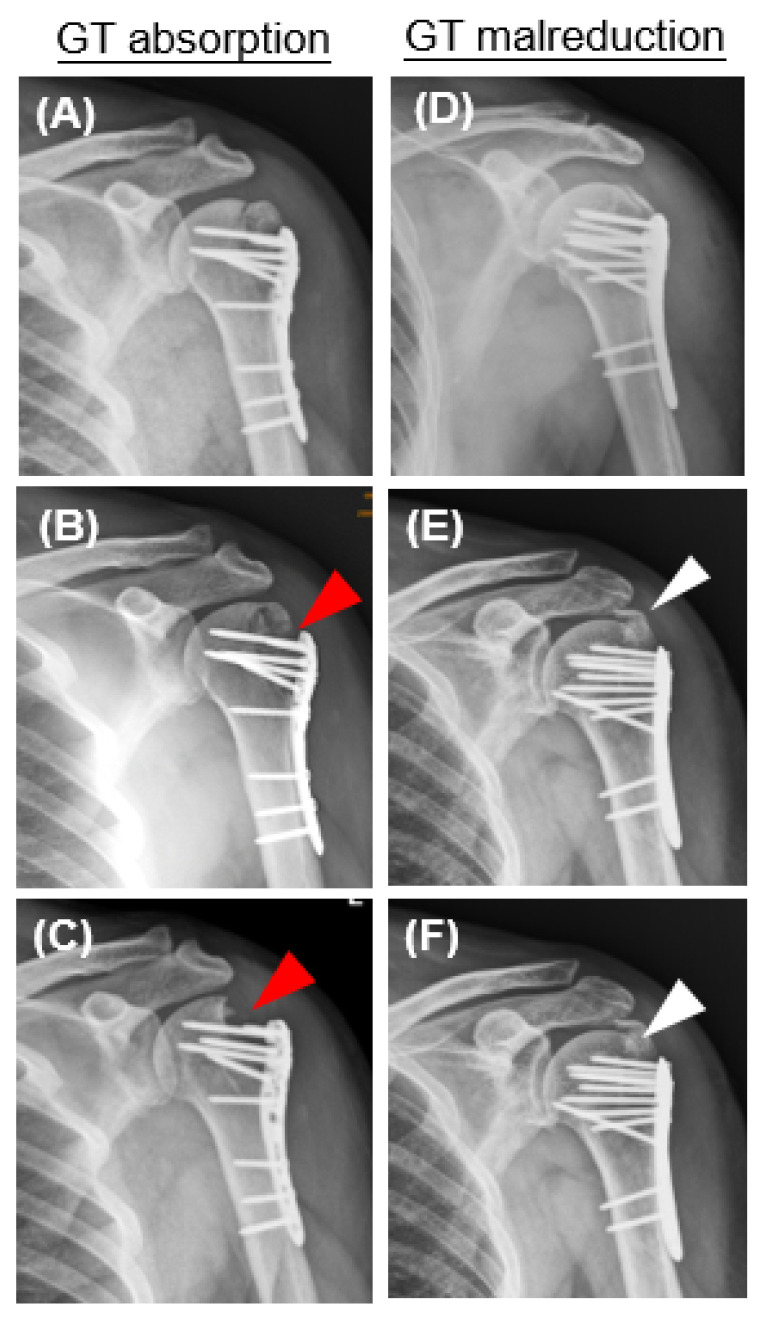
Plan radiographic evaluation of patient’s injured shoulder joint with greater tuberosity (GT) absorption or malreduction during follow-up period. (**A**–**C**) GT absorption was noted at 3 months postoperatively. The red arrowheads point out the radiolucent zone indicating GT absorption. (**D**–**F**) Postoperative plain radiographs of a patient presenting with GT malreduction. The GT fragment was well-reduced to its anatomical position in the initial postoperative image. (2D). At 1 month after surgery, GT was found displaced superiorly and posteriorly (2E) and remained at the malreduction position (white arrowheads) throughout the follow-up period (2F).

**Figure 3 diagnostics-13-02789-f003:**
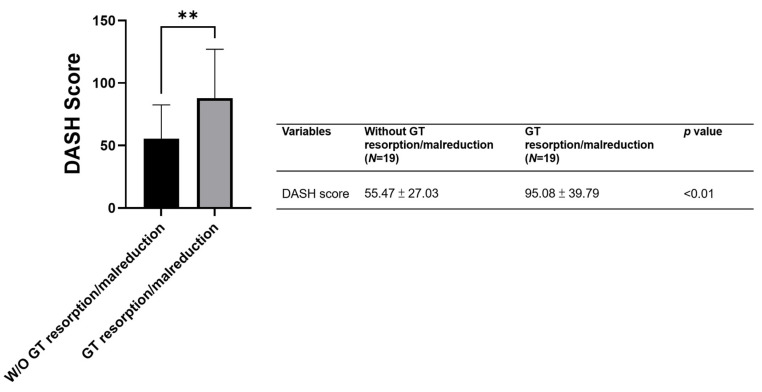
Postoperative functional outcome (DASH score) in patients with or without greater tuberosity resorption or malreduction. ** *p* < 0.01.

**Table 1 diagnostics-13-02789-t001:** Demographic and surgical characteristics of patients with proximal humeral fractures treated operatively.

Patients (*n* = 94)	Variables
Age (year)	63.32 ± 14.6
Gender	
Male	31; 33%
Female	63; 67%
Height(cm)	159.16 ± 9.54
Weight (Kg)	63.94 ± 14.41
BMI (Kg/m^2^)	25.09 ± 4.53
Alcohol consumption	9/94 (9.57%)
Smoking	8/94 (8.51%)
Deltoid tuberosity index	1.41 ± 0.16
Neer classification	
2-part	26/94 (27%)
3-part	60/94 (64%)
4-part	8/94 (9%)
AO/OTA classification	
11B1.1	82/94 (87%)
11C3	12/94 (13%)
Time to surgery (day)	2.93 ± 5.4
Time of follow up (weeks)	45.55 ± 25.62
Greater tuberosity resorption	13/94 (13.82%)
Malreduction of greater tuberosity	11/94 (11.7%)
Time from surgery to GT resorption or malreduction (week)	8.77 ± 7.86

BMI: body mass index.

**Table 2 diagnostics-13-02789-t002:** Demographic and surgical characteristics of matched patients with proximal humeral fractures with or without greater tuberosity resorption or malreduction. Values are presented as numbers or mean ± standard deviation.

Variables	W/O GT Resorption/ Malreduction (*n* = 19)	GT Resorption/Malreduction (*n* = 19)	*p* Value
Age (year)	70.58 ± 10.62	70.68 ± 11.04	0.9763
Male	4/19 (21%)	4/19 (21%)	0.99
Female	15/19 (79%)	15/19 (79%)	0.99
Height (cm)	157.42 ± 8.92	155.95 ± 7.71	0.58
Weight (Kg)	59.05 ± 9.16	67.16 ± 17.92	0.0876
BMI (Kg/m^2^)	23.85 ± 3.37	27.35 ± 6.03	0.0337
Smoking	0/19 (0%)	1/19 (5%)	0.229
Alcohol consumption	1/19 (5%)	1/19 (5%)	0.99
Deltoid tuberosity index	1.36 ± 0.15	1.41 ± 0.13	0.2261
Neer classification			
2-part	6/19 (32%)	6/19 (32%)	0.99
3-part	13/19 (68%)	11/19 (58%)	0.99
4-part	0/19 (0%)	1/19 (10%)	0.4762
AO/OTA classification			
11B1.1	18/19 (95%)	17/19 (89%)	0.99
11C3	1/19 (5%)	2/19 (11%)	0.99
Time to surgery (day)	4.11 ± 7.02	4.32 ± 8.45	0.5825
Time of follow up (week)	40.48 ± 26.22	42.23 ± 29.01	0.354
Artificial bone graft use	11/19 (58%)	12/19 (63%)	0.4657
Inadequate medial support	0/19 (0%)	3/19 (16%)	0.2297

**Table 3 diagnostics-13-02789-t003:** Comparison of postoperative radiographic parameters in patients with or without GT resorption or malreduction.

Variables	W/O GT Resorption/ Malreduction (*n* = 19)	GT Resorption/Malreduction (*n* = 19)	*p* Value
Lateral humeral offset (cm)	4.85 ± 0.44	4.56 ± 0.44	0.1939
Head diameter (cm)	4.71 ± 0.45	4.5 ± 0.34	0.2528
Head height (cm)	1.72 ± 0.2	1.68 ± 0.28	0.5957
Perpendicular height (cm)	4.95 ± 0.6	4.69 ± 0.61	0.2092
Neck-shaft angle (degree)	138.27 ± 13.73	137.82 ± 9.6	0.4765
Superior displacement of GT above articular surface	0/19 (0%)	13/19 (68.42%)	0.0463
Medial gap > 4 mm	0/19 (0%)	5/19 (26.32%)	0.0463
Use of calcar-specific screw	19/19 (100%)	16/19 (84.21%)	0.0001

## Data Availability

Full datasets are available upon reasonable request to the corresponding author.

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
