# Peer review of "Post-Operative Greater Tuberosity Resorption or Malreduction Is Associated with Poor Prognostic Outcomes in Patients with Proximal Humeral Fractures Treated Operatively—A Single-Center Retrospective Cohort Study"

_diagnostics, 2023, doi:10.3390/diagnostics13172789_

Round 1
Reviewer 1 Report (Previous Reviewer 2)
Dear authors,
Thank you for completing the necessary edits I said. Your research is eligible for publication in its current form.
Best
Reviewer 2 Report (Previous Reviewer 3)
The authors appropriately modified all of the reviewers' points. This study is worth publishing.
This manuscript is a resubmission of an earlier submission. The following is a list of the peer review reports and author responses from that submission.
Round 1
Reviewer 1 Report
1. Please provide an appropriate reference related to classification of greater tuberosity involvement based on plain film/CT in the methods section.
2. How was the sample size estimated for this study? Please provide the rationale.
3. What were the reasons for excluding 293 cases. Describe it in the results section.
4. How many investigators were involved in the reviewing of radiograph films? Ideally it should be done independently by at least 2 and consensus has to be built.
5. Please adhere to the concerned EQUATOR network guidelines pertaining to your study design and provide the concerned checklist.
6. Please state the exact p-values in Table 1 without alluding to n.s.
7. In the Table 1, under the Neer and AO/OTA classifications, only one p-value has to be there.
8. Please calculate the adjusted estimates based on confounding factors such as chronic illnesses.
Professional language editing is needed.
Author Response
Dear Respected Reviewers,
We deeply appreciate the useful comments and suggestions made by the reviewers on our manuscript. All suggestions have been given full consideration and changes made in the revised manuscript accordingly. Point-by-points responses are attached herein to facilitate the review of the revised manuscript.
Comments and Suggestions for Authors
Reviewer 1
- Please provide an appropriate reference related to classification of greater tuberosity involvement based on plain film/CT in the methods section.
Please refer to P.2, line 1097 where reference 19 has been added.
- How was the sample size estimated for this study? Please provide the rationale.
In present study, we retrospectively reviewed the clinical outcomes of patients diagnosed with proximal humeral fractures treated operatively from 2019-2021 and try to understand the cause-and-effect relationship and risks factors of the occurrence of GT malreduction or absorption. Because the current study was preliminary, we estimated the effect size of the final sample available instead of estimation of sample size required before study using a webtool available at https://sample-size.net/ (reference 1,2)
Reference
- Hulley SB, Cummings SR, Browner WS, Grady D, Newman TB. Designing clinical research : an epidemiologic approach. 4th ed. Philadelphia, PA: Lippincott Williams & Wilkins; 2013. Appendix 6A, page 73.
- Chow S-C, Shao J, Wang H. Sample size calculations in clinical research. 2nd ed. Boca Raton: Chapman & Hall/CRC; 2008. Section 3.2.1, page 58)
- What were the reasons for excluding 293 cases. Describe it in the results section.
Please refer to P.5, line 204-208 while the reasons for excluding the 293 cases have been added to the result section.
- How many investigators were involved in the reviewing of radiograph films? Ideally it should be done independently by at least 2 and consensus has to be built.
Please refer to P.2, line 103-105 which indicating that 3 independent orthopedic surgeons who were not involved in current study were involved in interpretation of radiographic films
- Please adhere to the concerned EQUATOR network guidelines pertaining to your study design and provide the concerned checklist.
We have fully adhered to EQUATOR network guidelines and attached the checklist.
- Please state the exact p-values in Table 1 without alluding to n.s.
We have made changes accordingly on Table 2 (not table 1). Please refer to P.6.
- In the Table 1, under the Neer and AO/OTA classifications, only one p-value has to be there.
We have made changes accordingly on Table 2.
- Please calculate the adjusted estimates based on confounding factors such as chronic illnesses.
Due to the relatively small sample size of the current study, further adjustment may be over adjusted and cause further loss of power. Thus, we keep the data unchanged in the revision.
Reviewer 2 Report
Dear authors,
I have studied your research in detail. Although your research is valuable regarding its subject, scope, and content, it requires some important revisions. In particular, you have styled your article very irregularly. Please revise the entire research by following the rules in the "instruction for authors" section of mdpi.
Below are the fixes I would like. I'll reevaluate your article after I've made the edits.
Abstract
-Please shorten the background part and add your purpose sentence to this part.
-Give your meaningful findings in the Results section. This part is not clear.
Introduction
-You have given this section in only two paragraphs, making it difficult to connect the topics. Please separate these sections from general to specific without breaking the topic integrity and make new additions if necessary.
-Please write the main hypothesis after your purpose statement.
Discussion
Please start this section by giving the major findings of your research, not general literature information. If necessary, add the major findings of your research at the beginning of this section with a new paragraph.
-Please do not forget to include your ethics committee approval number.
In the final revision, please completely revise the article according to the mdpi instructions for authors.
Yours sincerely
Minor editing of English language required
Author Response
Dear Respected Reviewers,
We deeply appreciate the useful comments and suggestions made by the reviewers on our manuscript. All suggestions have been given full consideration and changes made in the revised manuscript accordingly. Point-by-points responses are attached herein to facilitate the review of the revised manuscript.
I have studied your research in detail. Although your research is valuable regarding its subject, scope, and content, it requires some important revisions. In particular, you have styled your article very irregularly. Please revise the entire research by following the rules in the "instruction for authors" section of mdpi.
Below are the fixes I would like. I'll reevaluate your article after I've made the edits.
Abstract
-Please shorten the background part and add your purpose sentence to this part.
-Give your meaningful findings in the Results section. This part is not clear.
We have made changes accordingly. Please refer to P.6 & 7; line 226-260.
Introduction
-You have given this section in only two paragraphs, making it difficult to connect the topics. Please separate these sections from general to specific without breaking the topic integrity and make new additions if necessary.
We have made changes accordingly by dividing introduction section into 4 paragraphs. Please refer to P.2; line 57-98.
-Please write the main hypothesis after your purpose statement.
We have made changes accordingly. Please refer to P.2; line 96-98.
Discussion
Please start this section by giving the major findings of your research, not general literature information. If necessary, add the major findings of your research at the beginning of this section with a new paragraph.
-Please do not forget to include your ethics committee approval number.
Thank you! We have added the IRB approval number on P10.; line 387-389.
In the final revision, please completely revise the article according to the mdpi instructions for authors.
We have made changes accordingly.
Yours sincerely
Reviewer 3 Report
This study is interesting to study the functional outcome of proximal humerus fractures along with postoperative absorption or decreased abnormalities in tuberculosis.
The details of the review are as follows.
- The purpose of this study should also be described in Abstract.
- It should state what ORIF is an abbreviation of.
Author Response
Dear Respected Reviewers,
We deeply appreciate the useful comments and suggestions made by the reviewers on our manuscript. All suggestions have been given full consideration and changes made in the revised manuscript accordingly. Point-by-points responses are attached herein to facilitate the review of the revised manuscript.
This study is interesting to study the functional outcome of proximal humerus fractures along with postoperative absorption or decreased abnormalities in tuberculosis.
The details of the review are as follows.
- The purpose of this study should also be described in Abstract.
We have made changes accordingly. Please refer to P.2; line 40-42.
- It should state what ORIF is an abbreviation of.
We have made changes accordingly. Please refer to P.1; line 48